# Impact of the internet use on informal workers' wages: Evidence from China

Xiaofei Si[1], Maishou Li [1,2] *

1 School of Economics, Institute of Economics, Henan University, Kaifeng, Henan, China, 2 Institute of Rural Revitalization, Henan University, Kaifeng, Henan, China

* lms@henu.edu.cn

**Data Availability Statement:** All relevant data are within the manuscript and its Supporting information files. Please refer to suggested Data Availability Statements in section "China Family Panel Studies" at http://www.isss.pku.edu.cn/cfps/.

## Abstract

Based on the data from the China Family Panel Studies (CFPS) in 2018, the relationship between internet use and informal workers' wages and its internal mechanism were empirically discussed using the ordinary least squares and endogenous switching regression (ESR) model. The study found that internet use could significantly raise the level of wages of informal workers, and this conclusion still holds after the endogenous problem was solved through the endogenous switching regression model. Further research found that the influences of the internet use on informal workers' wages was heterogeneous. In other words, internet use has a more obvious impact on the wages of informal workers aged 31 to 40, 41 to 50, and 51 to 60 with the educational level of university and above in cities and towns, while it has a significant negative impact on wages of informal workers aged 16 to 20.

## Introduction

Today, informal employment is a common phenomenon in different countries of the world. Against the background of economic globalization, liberalization, global financial crisis, and the increasing demand for high-skilled labor in the formal sector, in recent years, the trend towards informal employment is becoming increasingly significant in many developed countries [1]. In 2018, the International Labour Organization (ILO) published a report, pointing out that currently, 2 billion workers in the world are engaged in informal employment and lack social security, labor rights, and decent working conditions, accounting for more than 61% of the labor force total population of the world [2]. Since the reform and opening-up, as the reform of China's economic system and state-owned enterprises continue to deepen, the scale of informal employment in China has also gradually increased, which has made important contributions to alleviating employment pressure, reducing poverty, and promoting economic development. However, compared with formal employment, informal employment which is also common in the labor market does not seem decent, and informal workers cannot enjoy the same benefits and treatment as formal workers. Most informal workers face poor working conditions, long working hours, unstable wages, and lack of social security [3, 4], and they are just "poor people who have a job" and are on the fringes of social differentiation [5]. Meanwhile, these workers have always been in a disadvantaged position in all labor relations.

**Funding:** The author(s) received no specific funding for this work.

**Competing interests:** The authors have declared that no competing interests exist.

**Abbreviations:** CFPS, stand for China Family Panel Studies; CNNC, stand for China Internet Network Information Center; ESR, stand for Endogenous Switching Regression; ISSS, stand for Institute of Social Science Survey; PSM, stand for Propensity Score Matching.

Even if they have entered the middle-income class, they still belong to the "fragile middle-income group". Therefore, it is imperative to increase the wage income of this population group and prevent polarization so that the Chinese population can avoid falling into the middle-income trap and achieve the goal of common prosperity for all people [6].

Besides, with the development and application of the Internet, artificial intelligence, and information technology, the labor market is likely to be tremendously impacted, especially on the wage income of employees, which mainly manifests itself through technology and information effects [7]. On the one hand, the substitution of low-skilled work by Internet technology will lead to a transfer of labor to more creative and productive sectors, increasing wage income while improving work efficiency. Considering the skill biased nature of technologies, the wage structure of education and experience is affected by the technological bias of human capital demand towards skilled labor that demonstrates efficient utilization of new technologies and help promote labor productivity [8, 9]. On the other hand, the fast developing Internet is conducive to information dissemination and communication, sparing job seekers the great costs of information acquisition [10]. Against this backdrop, informal workers can access more job recruitment information [11, 12], and find job opportunities with higher wages and better benefits [13], making it possible to improve remuneration levels, employment environment, and ultimately employment quality [14, 15]. Nevertheless, the Internet may stimulate substitution of labor and suppress wage increases. Specifically, on the one hand, the productivity increase brought about by technological progress means a decrease in labor demand for the same output. According to Schumpeter's theory of creative destruction, existing research results will inevitably be supplanted by higher levels of accomplishments in the future, and outdated research will produce negative externalities. Therefore, rapid technological progress is to diminish the current value of work, shorten the life cycle of jobs, and ultimately reduce labor demand. In addition, the increase in human capital prices with technological progress will further reduce corporate profits, leading to a negative impact on the initiative to provide creative jobs. On the other hand, the rise of the Internet and the information industry comes along with human-machine substitution. Technological paradigms show that technological diffusion can result in exacerbating structural unemployment. Data from the US labor market provides supportive evidence for this point of view: for every 0.1% increase in the number of robots, employment be down by 0.2% and wage levels by 0.42% [16]. In other words, the advancing robot technology has led to a decline in employment rates and wage levels.

As analyzed above, the relationship between Internet use and workers' income has been a research hotspot, albeit the lack of consensus on the impact of Internet use on workers' income. On the one hand, the widespread use of the Internet can lead to job creation, improved efficiency, and increased wages. On the other hand, it can cause unemployment among low- and middle-skill workers and affect their income levels. To date, Internet use and informal employment are omnipresent in China, arousing a keen research interest in the relationship between Internet use and informal workers' wages on the micro-level. With the Internet becoming indispensable in production and in daily life, how is informal workers' income affected by Internet use? Does such impact vary depending on the age, birthplace, and education level of this population group? Research on these issues can help understand the new forms and situations of employment in the Internet era to advance stability on the six fronts and security in the six areas, consolidate the bottom line of people's livelihood, enhance workers' income and social welfare levels, and ultimately achieve common prosperity. On this basis, in the age of rapid development of the internet, based on the data from China Family Panel Studies (CFPS) in 2018, the influences of the internet use on informal workers' wages and its internal mechanism were empirically discussed using the ordinary least squares (OLS) and endogenous switching regression (ESR) model, which provide a theoretical and practical

reference for proposing measures to raise the income levels of informal workers and to reduce the wage gap between formal and informal employment.

## Literature review and hypotheses

### Related literature on internet use and wage income

Researchers at home and abroad have done a lot of research on the relationship between the internet and workers' incomes. Studies have shown that internet use has a positive impact on personal income [17, 18]. In particular, wage income can be increased by 13.5% using the internet at work, and the return on income from internet use is higher especially in low-technology enterprises [19]. There are also studies on the impact of Internet use on the income of workers in different categories. The study by Mossberger, Tolbert, and Johns [20] discovered from the 2003 Current Population Survey data issued by the US Census Bureau that Internet use has a significant effect on the wage income of workers at all skill levels. Gao, Yan, and Bi [21] empirically analyzed the relationship between Internet use and wages using microdata and found that with Internet use, younger workers had an increased average hourly wage by 33%-76%. Some researchers studied the influence of simultaneous internet and computer use on the return on wages. The results showed that both were able to increase the income of workers, and the return on the income of the internet was 8% [22]. However, the paper mentioned above used the OLS model, so to a certain extent, there is the problem of sample selection. To avoid the above problem, researchers solved the sample selection problem using the propensity score matching model(PSM), and performed regression on wage earners and self-employed to solve the problem of sample selection due to the characteristics of their industry and enterprise. It was found that internet use could bring an income return of about 18% to 30% [23]. In China, PSM [24] and quantile regression [25] were applied to empirically examine the relationship between Internet use and workers' income. These studies have reached a conclusion that Internet use can increase worker income. However, some researchers argued that Internet use has no significant effect on workers' income. For instance, through statistical analysis of UK data, Bell found that Internet use had no significant influence on wage income [26] and could not increase the wages of Italian high school graduates [27]. In short, existing literature has demonstrated that Internet use influences on workers' income, and whether the influence is positive or negative is at the heart of our understanding of the relationship between Internet use and workers' income.

### Related literature on heterogeneity between internet use and wage income

Some researchers also pointed out that the influences of the internet use on workers' wages was heterogeneous. From the perspective of flexible employees, Qi, Ding, and Liu (2022) believed that internet use could significantly increase workers' wages, but such influences varied in different groups [28]. That is to say, compared with the middle-aged and the elderly, men and high-income groups, internet use can more significantly increase the wages of youth, women and low-income groups. From the perspective of rural residents, Liu, Zhang, and Yang (2021) believed that internet use can significantly increase the income of rural residents, but this is more significant for the young group and the highly educated group [29]. Yang and Zhou (2019), on the other hand, believed that internet use can more significantly increase the income of middle- and old-aged farmers with low educational levels [30]. From the perspective of female employment, Cao and Jiang (2020) believed that internet use can significantly increase the income of women and has a greater impact on the income of women who have not given birth [31]. From the perspective of different skills, Li (2019) believed that internet use has a bigger effect on income compensation of low-skilled groups [32]. Internet use has

increased the labor income of low-skilled groups significantly by 53.6%, but the influence on medium- and high-skilled groups is not significant. From the perspective of urban and rural areas, Jiang, Wang, Zhang, and Yue (2018) believed that internet use has different influences on the income of different resident groups [33]. That is, it has a relatively large impact on middle-aged people, rural residents, and highly educated groups. On the contrary, Hua (2018) believed that urban residents and those with a high level of education can receive higher return on income from internet use [34]. In summary, the influences of the internet use on wage levels across different groups is heterogeneous.

## Theoretical hypotheses

Based on the above analysis, this paper proposed research hypotheses H1 and H2:

**Hypothesis 1**: Internet use can help raise the wage level of informal workers; and

**Hypothesis 2**: The influences of the internet use on the wage level of informal workers are heterogeneous. Such influences vary according to place of household residence (urban or rural areas), age, and educational level.

# Empirical methodology and data

## Empirical methodology

**Baseline model.** Based on the Mincer's wage equation, this paper constructed the wage equation of informal workers with internet use as the core explaining variable, specified as follows:

$$Lnwage_i = \beta_0 + \beta_1 internet_i + \delta_j X_i + u_i \tag{1}$$

where the explained variable *Lnwage* is the logarithm of the hourly wage; the core explaining variable internet represents internet use; $i$ is the individual of informal workers; $X_i$ is a series of control variables, including individual level, family level, and regional level; $\beta_0$, $\beta_1$, and $\delta_i$ are the corresponding coefficients to be estimated, respectively; $\varepsilon_i$ represents the random error term.

**Endogenous switching regression model.** As mentioned above, by randomly specifying whether an informal worker uses the internet, based on the OLS regression in Eq (1), we can obtain an unbiased estimate of informal workers' wages. $\beta_1$ describes the influences of the internet on informal workers' wages. However, studies have shown that it is not random whether informal workers choose to use the internet, but the result of the comprehensive action of many unobservable factors. Such a result will affect the decision and wage levels of informal workers and lead to selective errors. While the measurement error due to missing variables can be eliminated using the general instrumental variable method, the heterogeneity of wages has not been taken into account. Although the influences of observable factors can be controlled using the propensity matching score method, it cannot handle the endogenous problems caused by unobservable factors, and the errors caused by the above unobservable factors can only be treated in the model using the ESR model [35]. On this basis, in this paper, the ESR model was used for calculation, specified as follows:

$$Internet_i^* = \theta_j C_{ij} + \xi_i \tag{2}$$

where $Internet_i^*$ is the latent variable of the virtual variable $Internet_i$. When $Internet_i^* > 0$, $Internet_i = 1$, indicating that informal workers use the internet. When $Internet_i^* \leq 0$, $Internet_i = 0$,

indicating that informal workers do not use the internet. $C_{ij}$ is a group of variables that affect whether informal workers choose to use the internet, and can be the same as the control variables in the benchmark regression model. However, to identify it, at least one variable in $C_{ij}$ should differ from that in the benchmark regression model, and this variable can directly affect the behavior of informal workers without affecting their wage levels. Drawing on the practices of Zhou and Liang (2018), Ding and Yuan (2019), and others, the popularization rate of the internet was selected as the identification variable [36, 37]. $\xi_i$ is the random error term, which is expected to be 0. θ is the corresponding coefficient.

Based on the above analysis, the equations of the wage level of informal workers in the two scenarios of the internet use and internet non-use are as follows:

$$Lnwage_{1i} = \delta_{1j}X_{1i} + \sigma_{1i}\lambda_{1i} + u_{1i} \quad \text{if } Internet_i = 1 \tag{3}$$

$$Lnwage_{0i} = \delta_{0j}X_{0i} + \sigma_{0i}\lambda_{0i} + u_{0i} \quad \text{if } Internet_0 = 0 \tag{4}$$

where $Lnwage_{1i}$ and $Lnwage_i$ represent the level of hourly wages of informal workers who use the internet and those who do not use the internet; $X_{1i}$ and $X_{0i}$ represent the factors affecting the wage levels of two types of informal workers; $u_{1i}$ and $u_{0i}$ represent the random disturbance term. To solve the problem of sample selection bias caused by unobservable factors, this study also introduced the inverse Mills ratio $\lambda_{1i}$ and $\lambda_{0i}$ and its covariance $\sigma_{1i} = cov(\xi_i, u_{1i})$ and $\sigma_{0i} = cov(\xi_i, u_{0i})$, and estimated the ESR model with full information maximum likelihood estimation method.

**Methods for estimating the processing effect of internet use on informal workers' wages.** The results of the ESR model reflect the influences of variables on the differences in wages between informal workers who use the internet and those who do not. To reflect the impacts of informal workers' decision on internet use on the wage levels in different scenarios, by constructing the counterfactual scenarios, this study compared the expected values of wages of informal workers who use the internet and those who do not use the internet in real scenarios and counterfactual hypothetical scenarios to estimate the average treatment effect of informal workers' decision on internet use.

Expected value of wages of informal workers who use the internet:

$$E(Lnwage_{1i}|Internet_i = 1) = \delta_{1j}X_{1i} + \sigma_{1i}\lambda_{1i} \tag{5}$$

Expected value of wages of informal workers who do not use the internet:

$$E(Lnwage_{0i}|Internet_i = 0) = \delta_{0j}X_{0i} + \sigma_{0i}\lambda_{0i} \tag{6}$$

At the same time, this study considered two counterfactual hypothetical scenarios that represent the expected value of wages of informal workers who actually use the internet when they do not use it:

$$E(Lnwage_{0i}|Internet_i = 1) = \delta_{0j}X_{1i} + \sigma_{0i}\lambda_{1i} \tag{7}$$

And the expected value of wages of informal workers who do not use the internet when they use it.

$$E(Lnwage_{1i}|Internet_i = 0) = \delta_{1j}X_{0i} + \sigma_{1i}\lambda_{0i} \tag{8}$$

From Eqs (5) and (7), the average treatment effect of wages of informal workers who use

the internet can be calculated as follows:

$$ATT = E(Lnwage_{1i}|Internet_i = 1) - E(Lnwage_{0i}|Internet_i = 1)$$

$$= \left(\delta_{1j} - \delta_{0j}\right)X_{1i} + (\sigma_{1i} - \sigma_{0i})\lambda_{1i} \qquad (9)$$

From Eqs (6) and (8), the average treatment effect of wages of informal workers who do not use the internet can be calculated as follows:

$$ATU = E(Lnwage_{1i}|Internet_i = 0) - E(Lnwage_{0i}|Internet_i = 0)$$

$$= \left(\delta_{1j} - \delta_{0j}\right)X_{0i} + (\sigma_{1i} - \sigma_{0i})\lambda_{0i} \qquad (10)$$

In this paper, the mean of $ATT$ and $ATU$ was used to estimate the average treatment effect of the internet use on wages of two types of informal workers.

## Variables

**Explained variable: Informal workers' wages.** First, there is currently no unified definition of informal employment in the academic community. Informal employment is generally defined worldwide as the form of employment in which there is an informal employment relationship and a lack of labor protection. On this basis, this paper drew on the practices of Lu and Zhang (2018) and other researchers [38], and regarded workers who have signed a labor contract in the process of employment and have endowment insurance and medical insurance as formal workers, and vice versa. Secondly, in addition to defining informal employment, drawing on the practices of Lu and Wang (2021) and other researchers, in this paper, hourly wage was selected as the proxy variable for informal workers' wages [39]. This is because compared with annual and monthly wages, using hourly wages can eliminate the influence of working hours on wages. Therefore, this study used the hourly wage of the main work in the CFPS database as the wage variable, that is, hourly wage = annual wage / (working hours per week × 4 × 12), and the log was treated.

**Treatment variable: Internet use.** According to the questions in the Internet module in the 2018 CFPS questionnaire about the internet, where the first question is "Do you access the Internet using mobile devices such as mobile phone and Tablet computers?" and the second is "Do you access the Internet using computers?", the respondent's answers of "No" to both questions indicate that the respondent does not use the internet, and the value is 0, while answering "Yes" to both indicate that the respondent uses the internet, and the value is 1.

**Identification variable: Popularization rate of the internet.** Drawing on the general practices of researchers, in this paper, the popularization rate of the internet was selected as the identification variable. This is because the popularization rate of the internet is the level of the internet development in a region and can directly affect whether informal workers use the internet. At the same time, as a macro factor, the popularization rate of the internet does not directly affect informal workers' wages.

**Other control variables.** Drawing on the general practices of researchers, in this paper, control variables, that is, individual characteristics, family characteristics, and regional characteristics, were added. Individual characteristic control variables include individuals' gender, age, marital status, years of education, place of household residence, union membership, state of health, and social status, while family characteristic control variables include the number of family members, and regional characteristic control variables include the eastern, central, western, and northeastern regions.

## Data

**Data source.** The data used in this paper are obtained from the database of adults of the China Family Panel Studies (CFPS: http://www.isss.pku.edu.cn/cfps/) conducted in 2018 by the Institute of Social Science Survey (ISSS) of Peking University. The purpose of the CFPS is to illustrate the changes in society, economy, population, education, and health by tracking and collecting data at the levels of individual, family, and community. Based on the research topic of this paper, the original samples were treated as follows: (1) Only the non-agricultural employed samples were retained; (2) Samples with working hours of 30 to 84 hours per week were retained [39]; (3) According to the mandatory age for retirement prescribed in China, the male samples aged 16 to 60 and the female samples aged 16 to 55 were retained; and (4) Samples of invalid questionnaires that had logical errors, in which key information was missing, and respondents refused to answer questions were eliminated. At the same time, considering the possible interference of outliers of wage income, this paper completed the ±1% Winsorization for hourly wage income and finally obtained 4,761 valid samples, including 3,551 samples that use the internet (accounting for 74.59%) and 1,210 samples that do not use the internet (accounting for 25.41%).

## Descriptive statistical analysis

The descriptive statistical results and the t-test of the main variables are presented in Table 1. Seen from the average wage, the average hourly wage of informal workers who do not use the internet is 2.08, while the average hourly wage of informal workers who use the internet is 2.35. Wages of informal workers who do not use the internet are lower than those of informal workers who use the internet, and the difference coefficient is significant at the significance level of 1%. In terms of age, the average age of informal workers who do not use the internet is

**Table 1. Variable definition and descriptive statistics.**

| Variables | Description of variables | Mean | | | Mean Diff |
|---|---|---|---|---|---|
| | | Overall (N = 4761) | Not using internet (N = 1210) | Use internet (N = 3551) | |
| Inwage | Hourly wages logarithm | 2.281 | 2.080 | 2.349 | -0.269*** |
| Gender | 1 = Male; 0 = Female | 0.615 | 0.677 | 0.594 | 0.082*** |
| Age | The actual age of the interviewee | 38.00 | 46.21 | 35.21 | 10.996*** |
| Age2 | Age squared divided by 100 | 15.65 | 22.17 | 13.43 | 8.734*** |
| Marriage | 1 = Married/ Cohabitation; 0 = Unmarried/Divorced/ Widowed | 0.757 | 0.898 | 0.709 | 0.190*** |
| Eduy | Years of education | 9.660 | 7.227 | 10.49 | -3.262*** |
| Register | Household registration: 1 = Agricultural; 0 = Non-agricultural | 0.747 | 0.815 | 0.724 | 0.091*** |
| Union_member | Union membership: 1 = Yes; 0 = No | 0.0689 | 0.0490 | 0.0760 | -0.027*** |
| Healthy | 1 = Poor; 2 = Fair; 3 = Good; 4 = Very good; 5 = Excellent | 3.256 | 3.070 | 3.319 | -0.249*** |
| Status | Social status level in area:1 = Very low;5 = Very high | 2.886 | 3.007 | 2.845 | 0.162*** |
| Family | Number of family members | 3.783 | 3.738 | 3.799 | -0.0610 |
| Region | 1 = Eastern; 2 = Central; 3 = Western; 4 = Northeast | 2.170 | 2.184 | 2.164 | 0.0200 |

*Note*:

* significant at 10%,

** significant at 5%,

*** significant at 1%.

***Data Source***: CFPS.

46.21 years old, while that of informal workers who use the internet is 35.21 years old. There is a big difference between the two, and the difference coefficient is significant at the significance level of 1%. From the viewpoint of gender, men account for 67.7% of informal workers who do not use the internet, while men account for 59.4% of informal workers who use the internet. The proportion of male informal workers who do not use the internet is higher than that of male informal workers who use the internet. From the marital status, married people account for 89.8% of the informal workers who do not use the internet on average and 70.9% of the informal workers who use the internet, both of which are dominated by married workers. Viewed in terms of years of education, the average years of education of informal workers who do not use the internet are 7.227 years, while the average years of education of informal workers who use the internet are 10.49 years. The difference between the two is 3.263 years. Seen from the place of household residence, the average share of people with rural area household registration among informal workers who do not use the internet is 81.5%, while the average share of those who use the internet is 72.4%. The proportion of the informal workers with rural area household registration who do not use the internet is higher than those who use the internet, and the difference coefficient is significant at the significance level of 1%. By union membership, 0.5% of informal workers who do not use the internet joined the trade union, while 7.6% of informal workers who use the internet joined the trade union. The difference between the two is significant. From the state of health, the average health level of informal workers who do not use the internet is lower than that of informal workers who use the internet, and both of the groups are relatively healthy. From the social status, the social status of informal workers who do not use the internet is higher than that of those who use the internet. From the number of family members, the number of family members of informal workers who do not use the internet is not significantly different from the number of those who use the internet. Regionally, the share of informal workers who do not use the internet in the eastern and central regions is lower than that of those who use the internet.

## Empirical results

### Baseline results

Based on the above analysis, the OLS method was used for the regression of Eq (1), and the results are shown in Table 2. Columns 1 is the impact of the internet use on informal workers' wages without the addition of control variables. The results show that internet use has a positive impact on informal workers' wages, and the regression coefficient is significant at the significance level of 1%. Columns 2 to 4 are the effects of Internet use on the wages of informal workers after gradually adding control variables such as individual characteristics, family characteristics, and regional characteristics. The results show that internet use still has a significant positive impact on informal workers' wages, confirming the research hypothesis H1 of this paper. This means that internet use can help to raise the wage levels of informal workers.

The analysis is carried out using columns 4 as an example. After controlling for all variables, internet use has a positive impact on informal workers' wages. This means that through the use of the internet, the wage levels of informal workers can be increased by 14.5%. Seen from the control variables, first, at the individual level, gender will affect the wage level of informal workers, that is, male informal workers earn higher wages than female workers, which can be attributed to the fact that male workers have to bear greater family financial responsibilities. In addition, the accumulation of work experience, human capital, life experience, and social networks also contributes to the increase in wage income. The age of informal workers will also affect their wage level, that is, there is an inverse-U relationship between age and wage levels. As informal workers grow older, their wage levels will initially rise, and when they reach a

**Table 2. Baseline regression for the impact of internet use on informality wages.**

| Variables | (1) | (2) | (3) | (4) |
|---|---|---|---|---|
| Internet | 0.269*** | 0.148*** | 0.148*** | 0.145*** |
|  | (0.028) | (0.031) | (0.031) | (0.031) |
| Gender |  | 0.389*** | 0.390*** | 0.397*** |
|  |  | (0.025) | (0.025) | (0.025) |
| Age |  | 0.069*** | 0.069*** | 0.071*** |
|  |  | (0.010) | (0.010) | (0.010) |
| Age2 |  | -0.089*** | -0.090*** | -0.093*** |
|  |  | (0.012) | (0.012) | (0.012) |
| Marriage |  | 0.167*** | 0.184*** | 0.175*** |
|  |  | (0.038) | (0.039) | (0.039) |
| Eduy |  | 0.041*** | 0.041*** | 0.041*** |
|  |  | (0.004) | (0.004) | (0.004) |
| Register |  | -0.161*** | -0.160*** | -0.169*** |
|  |  | (0.030) | (0.030) | (0.030) |
| Union_member |  | 0.463*** | 0.460*** | 0.460*** |
|  |  | (0.047) | (0.047) | (0.047) |
| Healthy |  | 0.052*** | 0.052*** | 0.051*** |
|  |  | (0.011) | (0.011) | (0.011) |
| Status |  | 0.051*** | 0.052*** | 0.054*** |
|  |  | (0.012) | (0.012) | (0.012) |
| Family |  |  | -0.014* | -0.014* |
|  |  |  | (0.008) | (0.008) |
| Middle |  |  |  | -0.153*** |
|  |  |  |  | (0.031) |
| West |  |  |  | -0.171*** |
|  |  |  |  | (0.030) |
| Northeast |  |  |  | -0.190*** |
|  |  |  |  | (0.039) |
| _cons | 2.080*** | -0.041 | -0.007 | 0.085 |
|  | (0.024) | (0.206) | (0.206) | (0.205) |
| N | 4761 | 4761 | 4761 | 4761 |
| $R^2$ | 0.017 | 0.162 | 0.162 | 0.170 |

*Note*: Standard errors in parentheses;

* significant at 10%,

** significant at 5%,

*** significant at 1%.

*Data Source*: CFPS.

certain age, their wage levels fall because informal workers have more energy and time to engage in informal work in order to earn higher wages when they are younger. When they reach a certain age, their physical functioning declines, and they can no longer devote time and energy to work, leading to a drop in their wage levels. Marital status also affects the wage levels of informal workers. Compared with other informal workers, the wage levels of married or cohabiting informal workers is higher because they have to shoulder more family responsibilities and spend more time at work to achieve higher wage levels. According to human capital theory, we know that years of education will affect the wage levels. Therefore, as the

educational level of informal workers increases, so will their wages. Compared with informal workers with urban area household registration, the wage levels of informal workers with rural area household registration is lower because the current strict dual-structure household registration system in China restricts workers' free movement in space and cannot help raise their wages. Union membership has a significant positive impact on wage levels of informal workers, that is, informal workers who join the trade union can achieve higher wage levels because a trade union can, to a certain extent, improve workers' negotiation capacity and effectively protect their legal rights and interests. The state of health will also have a significant positive impact on the wage levels of informal workers, that is, the better the state of health, the higher the wage levels of the informal workers. This is because as a human capital, health is a prerequisite for workers to be able to participate in work. Therefore, the better the state of health, the higher the wage levels. Secondly, at the family level, family size is negatively correlated with the wage levels of informal workers, with a larger family size meaning lower income. This is because it takes a greater amount of time to take care of a greater number of offspring, leading to reduced labor supply and lower income. Then, regionally, the region will have a significant negative impact on the wage levels of informal workers, that is, compared with the eastern region, the wage levels of informal workers in the central, western, and northeastern regions will decline by 0.153, 0.171, and 0.19, respectively.

## Analysis of regression results of endogenous switching regression model

The simultaneous estimation results of the decision on internet use and the wage levels of informal workers are shown in Table 3. From the Table, it can be seen that the independence assumption of the two-stage equation is rejected at the level of 1% in the LR test, which indicates that the decision equation and the wage impact equation of informal workers are correlated and the use of the ESR model is effective. $\rho_1$ is the correlation coefficient of the error term of the selection equation and the result equation of the internet use. $\rho_2$ is the correlation coefficient of the error term of the selection equation and the result equation of the internet non-use. $\rho_1$ and $\rho_2$ are significant at the level of 1%, which indicates that there is a selection deviation in the sample. Referring to the explanation of $\rho_1$ and $\rho_2$ of Lokshin and Sajaia (2004), the estimated value of $\rho_1$ is significantly negative, which indicates that the wage levels of informal workers using internet is higher than that of random individuals in the sample [40].

According to the results of influencing factors of the decision on internet use in Table 3, we know that gender, age, years of education, place of household residence, union membership, and social status are the key factors affecting whether informal workers use the internet among individual characteristics. In particular, compared with women, male informal workers are less likely to choose to use the internet, as most of the jobs of male informal workers are manual labor without internet use. Therefore, the probability of choosing to use the internet is relatively low. The age of informal workers will also affect the decision on internet use. With increasing age, the probability that informal workers decide to use the internet gradually decreases. The years of education of informal workers have a significant positive impact on the decision on internet use. Perhaps because informal workers with a high educational level have good learning skills and a high acceptance for new technologies, they tend to use the internet. The place of household residence has a significant negative impact on the decision on internet use. Because the internet infrastructure in rural areas is relatively backward compared with that in urban areas, and workers may not have access to internet devices such as computers, thereby reducing their likelihood to use the internet. Union membership will have a significant positive impact on the decision on internet use, while social status will have a significant negative impact on the decision on internet use. Among family characteristics, family size does not

**Table 3. Endogenous switching regression results of the impact of internet use on informal workers' wages.**

| Variables | Select equation | Result equation | |
|---|---|---|---|
| | Internet use | Use internet | Not using the internet |
| Gender | -0.103** | 0.395*** | 0.498*** |
| | (0.042) | (0.031) | (0.050) |
| Age | -0.052*** | 0.023*** | -0.005 |
| | (0.003) | (0.002) | (0.005) |
| Marriage | -0.011 | 0.179*** | 0.233*** |
| | (0.063) | (0.043) | (0.082) |
| Eduy | 0.073*** | 0.016*** | -0.009 |
| | (0.006) | (0.005) | (0.008) |
| Register | -0.171*** | -0.065* | -0.215*** |
| | (0.055) | (0.039) | (0.063) |
| Union_member | 0.149* | 0.369*** | 0.264** |
| | (0.089) | (0.063) | (0.107) |
| Healthy | 0.027 | 0.037*** | 0.046** |
| | (0.019) | (0.015) | (0.019) |
| Status | -0.037* | 0.074*** | 0.047** |
| | (0.020) | (0.016) | (0.021) |
| Family | 0.011 | -0.019* | 0.007 |
| | (0.014) | (0.010) | (0.016) |
| Middle | 0.339*** | -0.133*** | -0.253*** |
| | (0.070) | (0.039) | (0.059) |
| West | 0.255*** | -0.133*** | -0.229*** |
| | (0.077) | (0.039) | (0.058) |
| Northeast | 0.019 | -0.174*** | -0.134* |
| | (0.072) | (0.052) | (0.075) |
| Internet_pu | 0.020*** | | |
| | (0.004) | | |
| _cons | 0.957*** | 1.140*** | 1.647*** |
| | (0.288) | (0.129) | (0.303) |
| lns1 | | -0.043*** | |
| | | (0.014) | |
| $\rho_1$ | | -1.634*** | |
| | | (0.057) | |
| lns2 | | | -0.246*** |
| | | | (0.030) |
| $\rho_2$ | | | -0.269** |
| | | | (0.128) |
| LR | 334.36*** | | |
| N | 4761 | 3551 | 1210 |

*Note*: Standard errors in parentheses;

* significant at 10%,

** significant at 5%,

*** significant at 1%.

*Data Source*: CFPS.

have a significant influence on the decision on internet use, but it has a positive coefficient. Among the regional characteristics, the central and western regions have a significant positive impact on the decision on internet use, but the northeastern region does not have a significant positive impact on the decision on internet use. In terms of instrumental variables, the popularization rate of the internet is significant at the statistical level of 1%, and the estimation coefficient is positive. This indicates that the higher the popularization rate of the internet, the greater the possibility for informal workers to choose to use the internet.

From the estimated results of the result equation of the internet use in Table 3, we know that the regression coefficients of gender, marriage, union membership, state of health, and social status of informal workers are all positive among individual characteristics, indicating that whether internet is used have a positive impact on the wage levels of informal workers. The regression coefficients of the place of household residence of informal workers are all negative, indicating that whether they use the internet have a negative impact on their wage levels. Among family characteristics, family size has a negative impact on the wage levels of informal workers who use the internet, which indicates that it cannot help to raise their wage levels if the family size is too large, especially when there are too many minor children. In terms of regional characteristics, compared with the eastern region, informal workers in the central, western, and northeastern regions will have a significant negative impact on their wage levels regardless of whether they use the internet.

## Average treatment effects analysis

By using Eqs (9) and (10), the average treatment effect of the internet use and internet non-use on informal workers' wages can be further calculated. The results are shown in Table 4, where $y_{11}$ and $y_{00}$ correspond to Eqs (5) and (6), which represent the wage levels of informal workers when using the internet and not using the internet, respectively; $y_{01}$ and $y_{10}$ correspond to Eqs (7) and (8) and represent the counterfactual results, respectively; ATT and ATU represent the average processing effect of informal workers using the internet and not using the Internet, respectively. At the same time, to intuitively reflect the difference between the counterfactual and the real scenarios, this paper also drew the probability density diagram of informal workers' wages in different scenarios, as shown in Fig 1.

From Table 4, we can see that the average treatment effect of informal workers using the internet on their wage levels has a positive effect at a significant level of 1%. Considering the counterfactual assumptions, the ATT estimation results show that the wage levels of informal workers who have used the internet will decrease by 0.528 if they do not use the internet. The ATU estimation results show that if informal workers who do not use the internet can use the internet, their wage levels will rise by 1.62. This further illustrates that internet use can

**Table 4. The average treatment effect of internet use on informal workers' wages.**

| Types of informal workers | Use internet | | Not using internet | | ATT | ATU |
|---|---|---|---|---|---|---|
| *Informal workers using the Internet* | $y_{11}$ | 2.319 (0.006) | $y_{01}$ | 1.791 (0.005) | 0.528*** (0.008) | — |
| *Informal workers not using the internet* | $y_{10}$ | 3.701 (0.010) | $y_{00}$ | 2.081 (0.008) | — | 1.620** (0.013) |

*Note*: Standard errors in parentheses;

* significant at 10%,

** significant at 5%,

*** significant at 1%.

*Data Source*: CFPS.

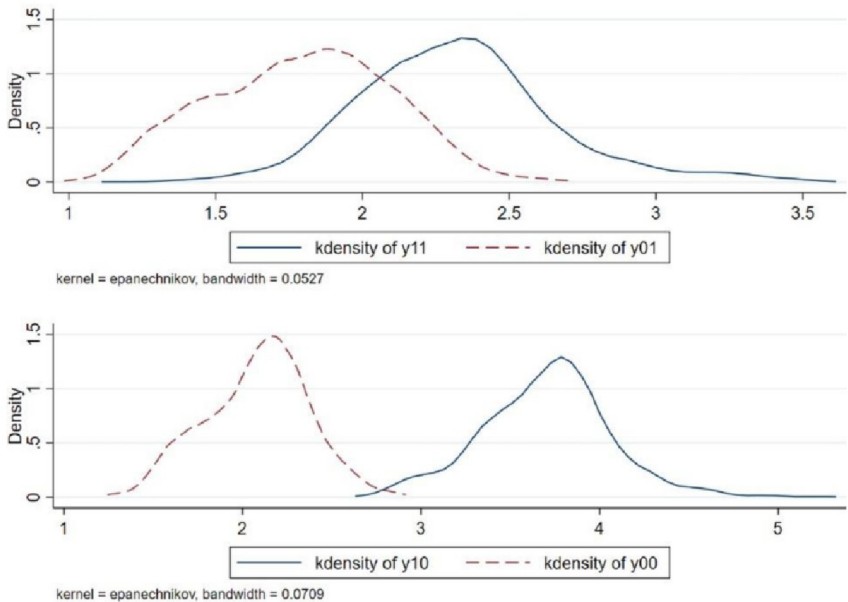

**Fig 1. Probability density of informal workers' wages in different scenarios.** *Data Source*: CFPS.

effectively raise the wage levels of informal workers, confirming the research hypothesis H1a of this paper.

## Robustness

To further verify the robustness of the analysis results, drawing on the practices of Peng and Wang (2021), hourly wage of the explained variable was replaced with total income [41]. And then, the estimation was carried out with OLS and ESR, respectively, and the results are shown in Table 5. According to the OLS regression results, we can know that internet use has a significant positive impact on the total income of informal workers, which is essentially consistent with the regression coefficient symbol and the significance level of the benchmark regression results. According to ESR's regression results, we can know that the popularization rate of the internet still has a significant positive impact on whether informal workers choose to use the Internet. In the LR test, the independence assumption of the two-stage equation is also rejected at the significance level of 1%, indicating that the decision equation and the wage impact equation of informal workers are correlated and the use of the ESR model is effective. $\rho_1$ is significant at the level of 1%, which indicates that there is selection bias in the sample and the results are essentially consistent with the results of the ESR above, further proving that the regression results are robust and reliable.

## Heterogeneity analysis

Due to differences in worker's age, place of household residence (urban or rural areas), and educational level, the use of the internet will lead to differences in wage levels among different groups of informal workers. On this basis, in this paper, informal workers were divided into groups by their age, place of household residence (urban or rural areas), and educational level to discuss the heterogeneity of the influences of the internet use on their wages.

**Table 5. Robustness test.**

| Variables | OLS | ESR | | |
|---|---|---|---|---|
| | lnincome | Result equation | | Select equation |
| | | Use internet | Not using internet | Internet Decisions |
| Internet | 0.163*** | — | — | — |
| | (0.028) | | | |
| Internet_pu | | | | 0.017*** |
| | | | | (0.004) |
| Control variables | YES | YES | YES | YES |
| _cons | 8.457*** | 9.542*** | 9.794*** | 1.039*** |
| | (0.194) | (0.120) | (0.282) | (0.302) |
| $\rho_1$ | | -1.423*** | | |
| | | (0.048) | | |
| $\rho_2$ | | | -0.201 | |
| | | | (0.132) | |
| N | 4761 | 3551 | 1210 | 4761 |
| $R^2$ | 0.157 | | | |
| LR | | 303.45 *** | | |

*Note*: Standard errors in parentheses;

* significant at 10%,

** significant at 5%,

*** significant at 1%.

*Data Source*: CFPS.

**Discussion based on age groups.** At different ages, internet use will have different impacts on the choice of informal employment by workers of different ages. To analyze in depth the impact of this difference, in this paper, the samples were divided into five groups by age, including between 16 and 20, between 21 and 30, between 31 and 40, between 41 and 50, and between 51 and 60, and were estimated by OLS. The regression results are shown in Table 6. We can see from Table 6 that internet use has a significant positive impact on the wages of informal workers aged 31 to 40, 41 to 50, and 51 to 60, a significant negative impact on the wages of informal workers aged 16 to 20, and no significant impact on the wages of informal workers aged 21 to 30.

**Discussion based on place of household residence (urban or rural areas).** With the further expansion of network coverage, the construction of network infrastructure in villages and cities has entered a period of rapid development, and the popularization rate of the internet has gradually increased. According to the 49th report of the China Internet Network Information Center (CNNC), the popularization rate of the internet in China has reached 73.0%, that in urban areas is 81.3%, and 57.6% in rural areas. There is still a big difference between urban and rural areas. In addition, in terms of the scale of the internet users, there is currently a total of 1.032 billion internet users in China, among which 748 million are urban internet users, accounting for 72.4% of the total number of the internet users, and 284.9 million are rural internet users, accounting for 27.6% of the total number of the internet users. There is still a huge difference in the number of the internet users between urban and rural areas. On this basis, in this paper, the samples were divided into urban and rural areas according to the type of respondents, and the regression results are shown in Table 7. We can see that internet use has a significant positive impact on the wage levels of rural and urban informal workers, but

**Table 6. Age group heterogeneity analysis results.**

| Variables | Age Group | | | | |
|---|---|---|---|---|---|
| | **16~20** | **21~30** | **31~40** | **41~50** | **51~60** |
| *Internet* | -0.606*** | 0.098 | 0.154** | 0.126*** | 0.277*** |
| | (0.199) | (0.093) | (0.067) | (0.045) | (0.065) |
| *Individual characteristics* | YES | YES | YES | YES | YES |
| *Family characteristics* | YES | YES | YES | YES | YES |
| *Regional characteristics* | YES | YES | YES | YES | YES |
| *_cons* | -1.785 | -5.458** | 1.465 | -2.167 | -6.574 |
| | (19.123) | (2.386) | (3.915) | (6.074) | (13.045) |
| *N* | 178 | 1286 | 1238 | 1326 | 733 |
| $R^2$ | 0.135 | 0.099 | 0.180 | 0.215 | 0.248 |

*Note*: Standard errors in parentheses;

* significant at 10%,

** significant at 5%,

*** significant at 1%.

*Data Source*: CFPS.

such effects differ. In other words, the influences of the internet use on the wages of urban informal workers is greater than that of rural informal workers.

**Discussion based on educational level.** Drawing on the common practices of researchers, the samples were divided into three groups according to educational level: junior high school and below, senior high school, and university and above. OLS is used for grouping regression, and the results are shown in Table 8. We can see from Table 8 that internet use has a positive impact on informal workers' wages with the educational levels of junior high school and below, senior high school, and university and above, and the most obvious impact is on informal workers with the educational level of university and above.

**Table 7. Analysis results of urban-rural heterogeneity.**

| Variables | Urban and rural grouping | |
|---|---|---|
| | **Rural** | **Urban** |
| *Internet* | 0.130*** | 0.150*** |
| | (0.046) | (0.041) |
| *Individual characteristics* | YES | YES |
| *Family characteristics* | YES | YES |
| *Regional characteristics* | YES | YES |
| *_cons* | 0.061 | 0.106 |
| | (0.311) | (0.276) |
| *N* | 2062 | 2699 |
| $R^2$ | 0.135 | 0.189 |

*Note*: Standard errors in parentheses;

* significant at 10%,

** significant at 5%,

*** significant at 1%.

*Data Source*: CFPS.

**Table 8. Educational heterogeneity analysis results.**

| Variables | Educational Level Grouping | | |
|---|---|---|---|
| | **Middle School or below** | **High School** | **College or above** |
| *Internet* | 0.130*** | 0.171* | 0.343** |
| | (0.033) | (0.094) | (0.154) |
| *Individual characteristics* | YES | YES | YES |
| *Family characteristics* | YES | YES | YES |
| *Regional characteristics* | YES | YES | YES |
| *_cons* | 0.721*** | 3.254** | -3.822*** |
| | (0.246) | (1.324) | (0.817) |
| *N* | 3079 | 899 | 783 |
| $R^2$ | 0.127 | 0.124 | 0.278 |

*Note*: Standard errors in parentheses;

* significant at 10%,

** significant at 5%,

*** significant at 1%.

*Data Source*: CFPS.

In summary, the sample groups are discussed according to age, place of household residence (urban or rural areas), and educational level. The influences of the internet use on the wage levels of informal workers in different groups is heterogeneous, which confirms the research hypothesis H2 proposed above.

## Conclusion and discussion

The rapid development of the internet, information and communication, and other technologies has a huge impact on the labor market, especially on wages of informal employment. On this basis, in the age of rapid development of the digital economy represented by the Internet, on the basis of data from the 2018 CFPS, OLS model, and ESR model were used to empirically test the influences of the internet use on informal workers' wages and its internal mechanism. In the studies, we find that internet use has a significant positive impact on the wages and employment of informal workers. Based on the counterfactual scenario, their wage levels will rise by 1.62 if informal workers who do not use the internet can use the internet. Secondly, the influences of the internet use on informal workers' wages is heterogeneous. From the age group, internet use has a significant positive impact on the wage levels of informal workers aged 31 to 40, 41 to 50, and 51 to 60, a significant negative impact on the wages of informal workers aged 16 to 20, and no significant impact on the wages of informal workers aged 21 to 30. Considered by the place of household residence (urban and rural areas), the influences of the internet use on the wage levels of urban informal workers is greater than that on rural informal workers. Differing by the educational level, the influences of the internet use on informal workers' wages with the educational level of university and above is greater than that on informal workers with the educational level of junior high school and below and senior high school.

Based on the research conclusion, this paper proposed that:

First, under the guidance of the policy of cyberpower, we will continue to promote the construction of the internet infrastructure and improve the basic capabilities of the network. In addition, in the process of expanding the investment and construction of digital infrastructure

such as internet information technology, we should continue to increase the popularization rate of the internet in China, reduce the digital gap between urban and rural areas and groups, optimize internet resources, and create a healthy and harmonious network environment for enterprises and workers. Second, we should promote the combination of the Internet, big data, and the labor market, build the labor market system based on the Internet and big data, promote the circulation of labor market information, and improve the asymmetry of labor market information. At the same time, we should encourage supply and demand parties of labor to publish job or recruitment information through internet platforms, continue to reduce the search cost and mismatch probability between employers and employees, and increase the opportunities for workers to get jobs. Third, we should promote the integrated development of information technology, digital technology, and the real economy, realize high-quality economic development, and provide workers with high-quality and efficient jobs.

## Supporting information

**S1 Data.**
(RAR)

## Author Contributions

**Conceptualization:** Xiaofei Si, Maishou Li.

**Data curation:** Xiaofei Si, Maishou Li.

**Formal analysis:** Xiaofei Si.

**Funding acquisition:** Maishou Li.

**Investigation:** Xiaofei Si.

**Methodology:** Xiaofei Si.

**Project administration:** Maishou Li.

**Resources:** Maishou Li.

**Software:** Xiaofei Si.

**Supervision:** Maishou Li.

**Validation:** Xiaofei Si.

**Visualization:** Xiaofei Si.

**Writing – original draft:** Xiaofei Si.

**Writing – review & editing:** Maishou Li.

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
