## [Decision Letter · Decision Letter 0]

19 Apr 2023

PONE-D-23-07656Impact of the internet use on informal workers’ wages: evidence from ChinaPLOS ONE

Dear Dr. Maishou Li

Thank you for submitting your manuscript to PLOS ONE. After careful consideration, we feel that it has merit but does not fully meet PLOS ONE’s publication criteria as it currently stands. Therefore, we invite you to submit a revised version of the manuscript that addresses the points raised during the review process.

We look forward to receiving your revised manuscript.

Kind regards,

Baljeet Singh, Phd

Academic Editor

PLOS ONE

Journal Requirements:

2. Please ensure that you have specified (1) whether consent was informed and (2) what type you obtained (for instance, written or verbal, and if verbal, how it was documented and witnessed). If your study included minors, state whether you obtained consent from parents or guardians. If the need for consent was waived by the ethics committee, please include this information.

Reviewers' comments:

Reviewer's Responses to Questions

**Comments to the Author**

1. Is the manuscript technically sound, and do the data support the conclusions?

Reviewer #1: Yes

Reviewer #2: Yes

2. Has the statistical analysis been performed appropriately and rigorously? 

Reviewer #1: Yes

Reviewer #2: Yes

3. Have the authors made all data underlying the findings in their manuscript fully available?

Reviewer #1: Yes

Reviewer #2: No

4. Is the manuscript presented in an intelligible fashion and written in standard English?

Reviewer #1: No

Reviewer #2: Yes

5. Review Comments to the Author

Reviewer #1: Impact of the internet use on informal workers’ wages:

evidence from China

Impact of the internet use on informal workers’ wages - evidence from China is interesting contribution.

I find the papers unique data set and sound methodology very impressive and economically logical. The authors have very succinctly modelled the available data and taken due considerations on various empirical issues.

However, I would like to rise some point that author would like to consider.

1. In the introduction section- the authors need to develop the theoretical/conceptual link between internet use and wages and need for this study. I know authors did talk on this in the second paragraph but in my view it’s not enough. Adding more insights on this will not only make a strong theoretical background of the paper but also give justification for need to undertake this kind of analysis.

2. On page 10 - interpretation of results. Check on the interpretation of variable gender. Was the wage variable in logs?

3. The authors also claim that there is gender wage discrimination based on the estimated coefficient of 0.397. I do not think you can claim such statements. Because differential of 39.7% could be due to gender related or factors associated with gender that you may have not controlled for in the regression. For instance, being male may be correlated with other productivity characteristics that have not being controlled for.

4. On page 11 – on family size. Authors justify negative impact of family size using ‘Many minor children’. Families in China can only have 2 kids which is normal with respect to replacement level of fertility so justifying this a lot is does not augur well with China’s child policy.

5. Avoid using ‘Seen from…” all the time to begin your sentences.

Reviewer #2: 1. Is the manuscript technically sound, and do the data support the conclusions?

The paper is well written and it is faithful to the hypotheses. The data has been sourced from a valid provider and sampled appropriately. The authors need to state the motivation for conducting the following research in the introduction section alongisde the hypothesis. Additionally, authors are recommended to ouline the major findings in the introduction for the readers.

2. Has the statistical analysis been performed appropriately and rigorously?

All the analysis and tests have been carried out to establish the research hypothesis.

3. Have the authors made all data underlying the findings in their manuscript fully available?

All the required findings are made available in the findings section. Authors are recommended to make the data available for replication.

4. Is the manuscript presented in an intelligible fashion and written in standard English?

The manuscript is well presented and clearly written.

6. PLOS authors have the option to publish the peer review history of their article (what does this mean?). If published, this will include your full peer review and any attached files.

Reviewer #1: No

Reviewer #2: No

---

## [Author Response · Author response to Decision Letter 0]

30 Apr 2023

Thank you to the Academic Editor and the Reviewers for the valuable suggestions provided for our manuscript. We have made every effort to revise the paper according to the suggestions. Please refer to the document 'Response to Reviewers' for details.

---

## [Editor Report · Decision Letter 1]

7 May 2023

Impact of the internet use on informal workers’ wages: evidence from China

PONE-D-23-07656R1

Dear Dr. Maishou Li

We’re pleased to inform you that your manuscript has been judged scientifically suitable for publication and will be formally accepted for publication once it meets all outstanding technical requirements.

Kind regards,

Baljeet Singh, Phd

Academic Editor

PLOS ONE

---

## [Editor Report · Acceptance letter]

10 May 2023

PONE-D-23-07656R1 

Impact of the internet use on informal workers’ wages: evidence from China 

Dear Dr. Li:

I'm pleased to inform you that your manuscript has been deemed suitable for publication in PLOS ONE. Congratulations! Your manuscript is now with our production department. 

Kind regards, 

on behalf of

Dr. Baljeet Singh 

Academic Editor

PLOS ONE